# Exome Sequencing Reveals Novel Germline Variants in Breast Cancer Patients in the Southernmost Region of Thailand

**DOI:** 10.3390/jpm13111587

**Published:** 2023-11-09

**Authors:** Panupong Sukpan, Surasak Sangkhathat, Hutcha Sriplung, Wison Laochareonsuk, Pongsakorn Choochuen, Nasuha Auseng, Weerawan Khoonjan, Rusta Salaeh, Kornchanok Thangnaphadol, Kasemsun Wanawanakorn, Kanyanatt Kanokwiroon

**Affiliations:** 1Department of Biomedical Sciences and Biomedical Engineering, Faculty of Medicine, Prince of Songkla University, Songkhla 90110, Thailand; panupong_2531@hotmail.com (P.S.); surasak.sa@psu.ac.th (S.S.); wison.l@psu.ac.th (W.L.); pongsakorn.c@psu.ac.th (P.C.); 2Medical Education Center, Naradhiwas Rajanagarindra Hospital, Narathiwat 96000, Thailand; nasuha_005@hotmail.com (N.A.); weerawun.k@gmail.com (W.K.); 3Translational Medicine Research Center, Faculty of Medicine, Prince of Songkla University, Songkhla 90110, Thailand; 4Department of Epidemiology, Faculty of Medicine, Prince of Songkla University, Songkhla 90110, Thailand; hutcha.s@psu.ac.th; 5Department of Surgery, Pattani Hospital, Pattani 94000, Thailand; rsalaeh@hotmail.com; 6Department of Surgery, Yala Regional Hospital, Yala 95000, Thailand; kornchanok_th@yahoo.com; 7Department of Primary Care, Su-ghai Kolok Hospital, Narathiwat 96120, Thailand; kssrcn3p@msn.com

**Keywords:** breast cancer, whole exome sequencing, germline variants

## Abstract

Germline carriers of pathogenic variants in cancer susceptibility genes are at an increased risk of breast cancer (BC). We characterized germline variants in a cohort of 151 patients diagnosed with epithelial BC in the southernmost region of Thailand, where the predominant ethnicity differs from that of the rest of the country. Whole exome sequencing was used to identify and subsequently filter variants present in 26 genes known to be associated with cancer predisposition. Of the 151 individuals assessed, 23, corresponding to 15.2% of the sample, exhibited the presence of one or more pathogenic or likely pathogenic variants associated with BC susceptibility. We identified novel germline truncating variants in *BRIP1*, *CHEK2*, *MSH6*, *PALB2*, and *PTEN* and annotated variants of uncertain significance (VUSs), both novel and previously documented. Therefore, it is advisable to use genetic testing as an additional risk screening method for BC in this area.

## 1. Introduction

Hereditary breast and ovarian cancer (HBOC) is recognized as an inheritable tumor syndrome. HBOC can often be traced to mutations in *BRCA1* and *BRCA2*, located on chromosomes 17 and 13, respectively [1]. Notably, not only *BRCA1/2* but also many other genes can harbor predisposing genetic variants that contribute to breast cancer (BC) susceptibility [2]. Individuals bearing pathogenic variants in *BRCA1* and *BRCA2* are at a substantially increased risk, of 65% and 45%, respectively, of developing BC. Under such risk susceptibility, the adoption of preventative measures such as prophylactic bilateral mastectomy and salpingo-oophorectomy has been shown to mitigate the likelihood of cancer [3]. Moreover, the implementation of Olaparib, a Poly(adenosine diphosphate-ribose) polymerase (PARP) inhibitor, has shown a noteworthy enhancement in the survival rate of individuals with metastatic BC, particularly those carrying pathogenic germline variants in the *BRCA* genes [4]. Patients with BC harboring homozygous or heterozygous germline mutations in the *ATM* gene exhibit heightened susceptibility to radiotherapy-induced toxicity subsequent to therapeutic interventions for BC. Furthermore, such genetic anomalies are associated with elevated susceptibility contralateral BC development [5,6]. Li-Fraumeni syndrome is an autosomal dominant genetic disorder resulting from the presence of a *TP53* mutation. This mutation is associated with an 85% elevated risk of developing BC and a significantly increased occurrence of subsequent primary tumors within the radiation-exposed area [7,8]. Moreover, individuals harboring pathogenic germline *CHEK2* mutations exhibit particularly elevated susceptibility to the onset of bilateral BC. Under such circumstances, the implementation of risk-reducing mastectomy is a beneficial intervention aimed at mitigating the risk of developing BC [9]. *PALB2* has a pivotal role as an interacting partner of *BRCA* within the homologous recombination double-strand break repair DNA pathway. This intricate involvement is significant, especially considering the high-risk implications associated with this BC-susceptibility gene. As such, prioritizing vigilant and timely monitoring for the early detection of this disease has become imperative [10].

More than 30 cancer-associated genes, ranging from low- to high-risk categories, are estimated to be involved in the development of BC [11]. The database www.omim.org (accessed on 1 July 2023) identifies 21 genes with significant associations with this malignancy. Germline genetic tests offer substantial advantages for individuals diagnosed with BC, as well as their family members and relatives. These tests yield high levels of satisfaction among the participants and demonstrate exceptional cost-effectiveness [12]. One study conducted in Norway showed that a panel test encompassing seven specific genes (*ATM*, *BARD1*, *BRIP1*, *CHEK2*, *NBN*, *RAD51C,* and *RAD51D*) demonstrates superior cost-effectiveness to singular *BRCA*-only testing [13]. Further, the guidelines established by the American Society of Breast Surgeons advocate for the necessity of genetic testing for individuals with a personal history of BC [14].

Since the advent of massively parallel sequencing technology, significant trends have emerged to expand genetic testing [15]. Although its value is apparent and recommended by various standard guidelines, germline genetic testing faces constraints in low- and middle-income economies such as Thailand [16]. The diversity of genetic germline variants and the prevalence of pathogenic alterations reflect ancestral lineage carrying sequence variants [17]. The southernmost expanse of Thailand, encompassing the Yala, Narathiwat, and Pattani provinces, is situated on the Malay Peninsula and has approximately 2 million inhabitants. Notably, 83% of the population in the area are Muslim-Thai, affiliating with the Malay Islam ethnicity in terms of race, language, and religious orientation [18]. Intriguingly, disparities in cancer incidence rates and BC subtypes between the Muslim-Thai and Buddhist-Thai subpopulations exert discernible influences on cancer-specific survival outcomes [19,20]. A previous investigation from central Thailand ascertained that 24% of patients with BC harbored germline pathogenic or reasonably pathogenic variants. The study also identified novel mutations in Thai individuals [21]. However, despite the distinctive ethnic composition, such data is conspicuously absent for the southernmost geographical regions.

In this study, we characterized germline variants in individuals with nonselective BC in the population in the deep south of Thailand using high-throughput sequencing. Through a comprehensive analysis of the sequence data, we meticulously annotated 26 BC susceptibility genes, namely, *ATM*, *BARD1*, *BRCA1*, *BRCA2*, *BRIP1*, *CASP8*, *CDH1*, *CDKN2A*, *CHEK2*, *EPCAM*, *HMMR*, *MLH1*, *MSH2*, *MSH6*, *MUTYH*, *NBN*, *NF1*, *PALB2*, *PHB1, PMS2*, *PTEN*, *RAD51C*, *RAD51D*, *RAD54L*, *STK11*, and *TP53,* spanning genes associated with low to high risk.

## 2. Materials and Methods

### 2.1. Population

We identified unselected cases with histologically confirmed diagnoses of epithelial BC or ductal carcinoma in situ (DCIS) from four general and regional hospitals located in the southernmost provinces of Thailand from January to June 2022. These accounted for a total of 151 cases from Naradhiwas Rajanagarindra Hospital, Pattani Hospital, Yala Regional Hospital, and Sungaikolok Hospital. The study protocol was approved by the Human Research Ethics Committee of the Naradhiwas Rajanagarindra Hospital (REC 001/2564). This study adhered to the guidelines outlined in the Declaration of Helsinki. For data collection, we procured approximately 10 mL of blood from each patient after pre-test counseling by a certified genetic counselor and obtaining written informed consent. The specimens were stored in EDTA tubes. These samples were meticulously preserved in the Central Research Laboratory and housed in the Translational Medicine Research Center of the Faculty of Medicine, Prince of Songkla University, in accordance with ISO 15189:2012 and ISO 15190:2020 standards [22,23].

### 2.2. Whole Exome Sequencing (WES)

Genomic DNA was extracted from the buffy coat using the High-Pure PCR Template Preparation Kit (Roche, Berlin, Germany). The quality and quantity of the obtained DNA were assessed using a bioanalyzer (Agilent Technologies, Santa Clara, CA, USA) and a Nanodrop (Thermo Scientific, Waltham, MA, USA). The Agilent SureSelect XT V6 kit (Agilent) was used to capture and enhance the exome regions. The Illumina Next-Seq 550 system (Illumina, San Diego, CA, USA) was used to perform paired-end exome sequencing and paired-end targeted resequencing with a read length of 150 bp. The average reading depth was approximately 200X, resulting in an output of approximately 9 gigabytes per sample.

### 2.3. Bioinformatics Analysis

Quality control of the FASTQ files was conducted using FastQC, followed by trimming using Trimmomatic-0.38. Subsequently, we performed another round of quality assessment. Satisfactory data were aligned against the reference genome (GRCh38 assembly) using BWA-0.7.17. The resulting alignments were then converted into BAM files using Samtools-1.11. Duplicate reads were sorted and removed from the BAM file using Picard. We further enhanced the data accessibility by indexing it using Samtools-1.11. For refinement, we recalibrated the data and identified the variants using GATK-4.1.2.0. The resulting variants were annotated using the SnpEff & SnpSift. Subsequently, we filtered the VCF file to include only variants with moderate or strong effects on the 26 BC susceptibility genes using RStudio. To obtain novel insights, we analyzed previously unreported single nucleotide polymorphisms (SNPs) using sorting intolerant from tolerant (SIFT) and PolyPhen.

The presence of variants was validated using Sanger sequencing. Variants deemed reportable were classified as pathogenic variants (PVs), likely pathogenic variants (LPs), or variants of uncertain significance (VUSs) following the 2015 guidelines of the American College of Medical Genetics and Genomics (ACMG) [24]. This classification was reinforced by cross-referring to Vasome and ClinVar, both of which were accessed on 1 July 2023. This study did not encompass an evaluation of substantial, large genomic rearrangements or the analysis of copy number variants.

### 2.4. Sanger Sequencing

Specific primers for germline variants were designed using Primer3Plus (Boston, MA, USA) (Appendix A) [25]. Target DNA was amplified by PCR using a T100 thermal cycler (Bio-Rad, Hercules, CA, USA). After amplification, DNA was qualified, and the samples were electrophoresed on an agarose gel to assess their size and purity. Subsequently, the PCR products were sequenced using an Applied Biosystems 3730/3730xl DNA Analyzer (Thermo Scientific, CA, USA) and the Big Dye Terminator v3.1 Cycle Sequencing Kit (Applied Biosystems, CA, USA). Data were analyzed using the Sequencing Analysis Software v7.0.

## 3. Results

### 3.1. Patient Characteristics 

We included 151 women diagnosed with DCIS or epithelial BC aged 23–79 years of age (mean age = 48.3 years); the patient characteristics are presented in Table 1
Appendix A. Of the 151 patients included, 91 (60.3%) were younger than 50 years, and 60 (39.7%) were older than 50 years. Further, 131 (86.8%) had invasive ductal carcinoma, 6 (4%) had DCIS, and 14 (9.3%) had other pathological types of cancers. Sixty-one (40.4%) patients had Luminal A subtype, 32 (21.2%) had Luminal B, 30 (19.9%) had HER2-enriched, and 28 (18.5%) had triple-negative BC (TNBC). Approximately half of the patients (78, 51.7%) were diagnosed with late-stage (stage III or IV) BC. Individuals afflicted with BC and diagnosed at 50 years or younger exhibited a considerably higher prevalence of PVs or LPs than their counterparts aged 50 years or older (*p* < 0.05). Of note, among the 28 patients with the TNBC molecular subtype, 32.1% were identified as carriers of PVs or LPs of BC-susceptibility genes. This prevalence is in stark contrast to the lower prevalence of such mutations in other BC subtypes, with rates of 11.5% in Luminal A, 12.5% in Luminal B, and 10.0% in HER2-enriched subtypes. Importantly, these statistical differences were substantiated based on a *p*-value of 0.05. There was no statistically significant difference in the proportion of germline mutations with respect to the pathological group, stage at diagnosis, or religion.

### 3.2. Germline Variant

Among the 26 genes on the list, PVs or LPs were identified in 11 genes: *ATM*, *BRCA1*, *BRCA2*, *BRIP1*, *CHEK2*, *MSH6*, *NF1*, *PALB2*, *PTEN*, *RAD51D*, and *TP53* (Table 2) (Appendix A). *BRCA2* has the highest number of variants, followed by *BRCA1*. Figure 1 shows the distribution of PVs and LPs variants in this study. Twenty-three of the 151 patients with BC (15.2%) carried at least one PV/LP in a gene known to be associated with an increased cancer risk, including a 48-year-old woman with mutations in both *ATM* (c.2377-2A>G (Splicing)) and *BRCA2* (c.7558C>T (p.Arg2520Ter)). Two unrelated patients with BC diagnosed with TNBC at 28 and 51 years old carried *large deletion* mutations in *BRCA1* c.1863_1885del (p.His621GlnfsTer7). We previously published preliminary information on this in 2023 [26]. Furthermore, we detected novel PVs and LPs in *BRIP1, CHEK2, MSH6, PALB2*, and *PTEN* (Table 2) (Appendix A). Moreover, according to the ACMG criteria, VUSs are detected in 15 of the 26 BC-related genes assessed (Table 3) (Appendix A). Five individuals diagnosed with BC possessed a VUS located in the *MUTYH* c.892-2A>G (splicing) region, specifically, the rs77542170 locus. These cases could be further classified into distinct molecular subtypes associated with the expression of HER2 in BC. Four of the individuals had the HER2-enriched subtype, whereas one case was classified as Luminal B.

## 4. Discussion

Cancer is a genetic disorder that originates in singular cells; cumulative abnormalities in cellular genetics lead to cancer cells that undergo uncontrollable growth and divisions [27]. Somatic loss of both alleles of a tumor suppressor gene causes protein dysfunction, resulting in the loss of cellular regulatory function [28,29]. At the germline level, the inherited first hit occurs as a heterozygous lesion, increasing the risk of developing the second hit at the organ level (somatic mutation) [30,31]. Individuals carrying a germline variant are at an increased risk of developing cancer. Information on the genetic risk of an individual is useful for personal risk identification, not only for the affected individual but also for the preventive management of family members [32].

We performed WES in 151 women diagnosed with BC using high-throughput sequencing. We found that 15.2% of patients with BC carried a PV or LP. Notably, the most prevalent variants were annotated in *BRCA1/2* genes, which are the main markers of inherited BC. Interestingly, the frequency of mutations was consistent with a previous study in the Taiwanese population, despite differences in the inclusion criteria in the two studies; in the previous study, participants were included based on hereditary risk and reported a 13.5% carrier rate of pathogenic germline mutations [33]. Furthermore, a large study encompassing 8085 unselected Chinese BC patients reported a germline mutation rate of 9.2% [34]. Another study conducted in Sarawak, Malaysia, analyzed blood DNA obtained from 467 patients with BC and revealed pathogenic variants in *BRCA1* and *BRACA2* in 2.8% and 3.2% of patients, respectively [35]. In our analytical studies, we did not identify any BC-related genes within the PV, LP, or VUS categories among the seven genes, namely *CASP8*, *CDKN2A*, *EPCAM*, *HMMR*, *NBN*, *PHB1,* and *STK11*. This dataset is advantageous for the prospective development of a multigene panel test. In the context of our investigation, we have discerned the presence of PVs, LPs, or VUSs of the genes *ATM, BARD1, BRCA1, BRCA2, BRIP1, CDH1, CHEK2, MLH1, MSH2, MSH6, MUTYH, NF1, PALB2, PMS2, PTEN, RAD51C, RAD51D, RAD54L, STK11,* and *TP53* among BC patients residing in the southernmost region of Thailand. Consequently, the utilization of the *BRCA* genetic test cannot be considered a comprehensive diagnostic approach for this particular population.

We previously investigated the impact of germline *BRCA1* mutations on BC risk, which is known to increase the probability of BC development by 45–80%. Furthermore, these mutations are often associated with TNBC, a subtype associated with low survival rates [36,37]. In this study, we discerned the presence of PV or LP mutations within the *BRCA1* gene in four cases, constituting 2% of the total samples. Notably, three of these four cases were associated with development of the TNBC molecular subtype. One study conducted cancer multigene panel tests on 296 female BC patients from the Malwa region of Punjab, India. The results of these tests did not reveal the presence of a *BRCA1* germline mutation [38]. In a study of patients from central Thailand with a diagnosis of BC and classified as having a high risk of HBOC according to the 2019 National Comprehensive Cancer Network (NCCN) guidelines, 29 of the total 309 individuals (9.5% of the patients) harbored germline *BRCA1* mutations [21].

Another gene of interest is *BRIP1*, in which PVs are associated with low penetrance BC [39]. Moreover, variants in *BRIP1* are also linked to an increased risk of hereditary ovarian cancer [40]. In our study, we identified a new PV in *BRIP1* at position c.195_196insC (p.Ser66fs) in a patient diagnosed with BC at the age of 40.

A previous investigation in a Malaysian population identified an uncommon germline variant in *CHEK2* [41]. We also identified two variants, a PV and a VUS, in *CHEK2*, of which the VUS is novel.

In the present study, we also identified a novel loss-of-function variant, *MSH6* c.4003del (p.Glu1335LysfsTer11), highlighting its potential relevance to BC predisposition. *MSH6* mutations have been implicated in Lynch Syndrome, a condition associated with an elevated risk of colorectal and other cancers [42]. The partner and localizer of *BRCA2* (*PALB2*) is a crucial player in the DNA damage response and is recognized as a BC predisposition gene [10]. A previous study showed that loss-of-function mutation carriers in *PALB2* increases the risk of BC by 9.5 times, compared to the non-carriers [43]; *PALB2* mutations were detected in 2 out of 151 cases. We also identified a novel *PALB2* variant, c.2101delT (p.Ser701fs), which is associated with BC susceptibility. An analysis of the populations in Malaysia and Singapore revealed a limited occurrence of *PALB2* mutations (2/122) among patients with BC with a family history of BC, indicating the rarity of these variants [44].

We also detected a novel LP in the *PTEN* gene at position c.154_154+1 insGG, resulting in the mutation p.His53GlyfsTer10. This variant is associated with Luminal A BC. Recent studies have highlighted the presence of *HER2*-overexpressing carcinomas [45]. Furthermore, the identification of a detectable mutation in the *PTEN* gene has been shown to increase the risk of BC by 85% [46]. Consequently, surveillance measures in *PTEN* variant carriers are advised to be at the same level as those recommended for individuals carrying mutations in the *BRCA1/2* genes [47].

We analyzed PVs and LPs in central Thailand and identified disparities in the spectrum of variants compared with previous studies [21,48]. In a study of 389 patients with cancer which fulfilled the hereditary cancer criteria and who underwent multigene panel testing, only a PV in TP53 c.1024C>T(p.Arg342Ter) had a similar mutation to our study. One study found a divergent variant spectrum within a cohort of 12 individuals who were carriers of PV/LP *BRCA1/2*. Twenty-six of the 112 patients diagnosed with non-mucinous epithelial ovarian cancer differed in the truncating variant spectrum compared to the findings reported in this study [49].

VUSs lie in a complex realm, straddling the boundaries between likely benign and likely pathogenic categories [24]. In this study, variants in this category were identified in 15 genes, as highlighted in Table 3
Appendix A. Notably, a considerable portion of VUS instances undergo reclassification, gravitating towards likely benign or benign variant classifications [50,51]. In this study, we identified a VUS in *MUTYH* c.892-2A>G (splicing) (rs77542170) in 5 patients with BC. It is imperative to note that biallelic mutations in *MUTYH* are correlated with the onset of familial adenomatous polyposis-2 and colorectal cancer. Furthermore, individuals carrying monoallelic *MUTYH* mutations exhibit an augmented susceptibility to a spectrum of cancers, including gastric cancer, liver cancer, colorectal cancer, and endometrial cancer [52,53]. A pertinent case-control investigation conducted on the Jewish population in North Africa revealed a statistically significant association between the G396D monoallelic mutation in *MUTYH* and an elevated risk of BC (OR:1.86, 95% confidence interval:1.02–3.39; *p* < 0.039) [54]. A comprehensive study encompassing 225 instances of monoallelic *MUTYH* mutations originating in the USA, Australia, New Zealand, and Canada yielded crucial insights. The calculated cumulative risks of developing BC in females by the age of 70 stands at 11% (95% CI:8.3–16), signifying a subtle but discernible elevation in the likelihood of BC occurrence in carriers of *MUTYH* mutations [55]. Furthermore, an Italian Multicenter Study involving 560 men with BC and 1540 control participants performed exhaustive next-generation sequencing of 50 genes associated with cancer susceptibility, including *MUTYH,* and found biallelic or monoallelic pathogenic variants in *MUTYH* to be intricately linked to an augmented vulnerability to BC in men [56]. Additional investigations concerning the mono-allelic *MUTYH* c.892-2A>G (splicing) rs77542170 variant in the context of BC must be undertaken, specifically for our designated population.

The primary limitation of our study pertains to the relatively modest sample size, comprising only 151 cases. Thailand is a middle-income nation, and the availability of genetic testing within its borders remains limited. Further, as a result of the constraints related to our data-collection procedures, we were unable to conduct a follow-up analysis of treatments for the purpose of monitoring and assessing survival rates. WES is not the optimal assay for the detection of substantial genomic rearrangements, including large deletions, insertions, or copy number variations. Moreover, we did not employ alternative methodologies to address this issue; however, we anticipate that this aspect will be investigated in future research endeavors. With regard to the intron region, a non-protein coding region, we must recognize that mutations occurring within this domain can influence gene expression and potentially initiate carcinogenic processes. Nevertheless, it is imperative to emphasize that an in-depth exploration of this specific region has not been undertaken thus far. The intron represents an intriguing avenue of investigation, warranting further scrutiny in prospective research endeavors.

For the analysis of VUSs, following the ACMG guidelines, we have observed a classification pattern denoting a gray color variant. This categorization poses a challenge in terms of providing a definitive assignment as either an LP or a likely benign variant. The pursuit of additional research endeavors for the further categorization of VUSs could bolster the available data and provide valuable support for the reclassification of variants. In reclassifying VUSs, it is crucial to consider that the existing data provide substantial support of the observation that a noteworthy percentage of these variants generally undergo reclassification into lower pathogenicity categories [57].

## 5. Conclusions

In conclusion, we showed that 15.2% of unselected patients with BC from the southernmost region of Thailand showed potential PV/LP germline variants, which demonstrates their increased susceptibility to BC. Moreover, we identified novel PVs, LPs, and VUSs in cancer-predisposing genes using the criteria outlined in the American College of Medical Genetics and Genomics guidelines. Determining the definitive link between these variants and BC necessitates further detailed investigation. Our study highlights BC susceptibility in this unique population. We suggest that genetic testing should be considered as an additional risk screening method for BC in this region for early detection and prevention strategies.

## Figures and Tables

**Figure 1 jpm-13-01587-f001:**
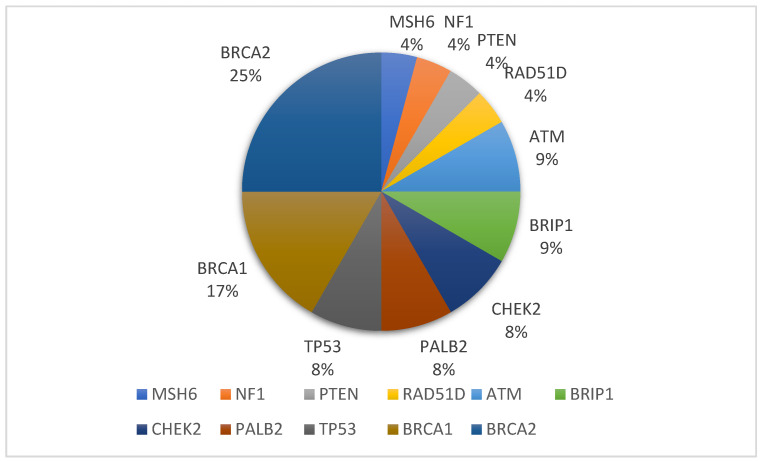
Distribution of the 11 breast cancer susceptibility genes among the 26 annotated genes. Importantly, the genes excluded from the presentation (*BARD1*, *CASP8*, *CDH1*, *CDKN2A*, *EPCAM*, *HMMR*, *MLH1*, *MSH2*, *MUTYH*, *NBN*, *PHB1*, *PMS2*, *RAD51C*, *RAD54L*, and *STK11*) were not associated with identified PVs or LPs within the study cohort. PV: pathogenic variant; LP: likely pathogenic variant.

**Table 1 jpm-13-01587-t001:** Patient characteristics.

Study Characteristic (151 Cases)	No (%)	No. among the 23 Cases of PVs/LPs (%)	*p*-Value
Age at diagnosis (year)			0.02
≤50	91 (60.3)	19 (20.9)	
>50	60 (39.7)	4 (6.7)	
Pathological group			0.13
Invasive ductal carcinoma (IDC)	131 (86.8)	21 (16.0)	
Ductal carcinoma in situ (DCIS)	6 (4.0)	2 (33.3)	
Others	14 (9.3)	0 (0.0)	
Molecular subtypes			0.05
Luminal A	61 (40.4)	7 (11.5)	
Luminal B	32 (21.2)	4 (12.5)	
HER2-enriched	30 (19.9)	3 (10.0)	
TNBC	28 (18.5)	9 (32.1)	
Stage at diagnosis			0.48
DCIS	6 (4.0)	2 (33.3)	
Stage I	18 (11.9)	4 (22.2)	
Stage II	49 (32.5)	4 (8.2)	
Stage III	53 (35.1)	8 (15.1)	
Stage VI	25 (16.6)	5 (20.0)	
Religion			0.16
Muslim-Thai	106 (70.2)	19 (17.9)	
Buddhist-Thai	45 (29.8)	4 (8.9)	

Breast cancer staging followed the American Joint Committee on Cancer classification (7th edition). PVs: Pathogenic variants/LPs: Likely Pathogenic variants. *p*-value for the proportion of LPs/PVs within each category.

**Table 2 jpm-13-01587-t002:** Result of likely pathogenic or pathogenic germline variant in 23 breast cancer patients.

Gene	HGVS	No. of Carriers	Age at Diagnosis (Years)	dbSNP ID	Molecular Subtype
*ATM*	c.8373C>A (p.Tyr2791Ter)	1	36	rs1060504292	Luminal B
	c.2377-2A>G (p.Splicing)	1	48	rs1057516553	Luminal A
*BRCA1*	c.2059C>T (p.Gln689Ter)	1	54	rs273898674	TNBC
	c.1863_1885del (p.His621GlnfsTer7)	2	51, 28	-	TNBC
	c.5068A>C (p.Lys1690Gln)	1	48	rs397507239	HER2-enriched
*BRCA2*	c.6352_6353del (p.Val2118LysfsTer10)	1	48	rs80359576	TNBC
	c.5164_5165del (p.Ser1722TyrfsTer4)	1	39	rs80359490	TNBC
	c.7558C>T (p.Arg2520Ter)	1	48	rs80358981	Luminal A
	c.3680_3681del (p.Leu1227GlnfsTer5)	1	48	rs80359395	Luminal A
	c.3283C>T (p.Gln1095Ter)	1	48	rs397507662	Luminal A
	c.262_263del (p.Leu88AlafsTer12)	1	30	rs276174825	TNBC
*BRIP1*	c.2990_2993del (p.Thr997ArgfsTer61)	1	39	rs771028677	Luminal B
	c.195_196insC (p.Ser66fs)	1	40	Novel	HER2-enriched
*CHEK2*	c.1178C>G (p.Pro393Arg)	1	57	Novel	Luminal B
	c.246_260del (p.Asp82_Glu86del)	1	47	rs587780181	HER2-enriched
*MSH6*	c.4003del (p.Glu1335LysfsTer11)	1	44	Novel	TNBC
*NF1*	c.6592_6593del (p.Asp2198fs)	1	40	rs765010702	Luminal B
*PALB2*	c.2257C>T (p.Arg753Ter)	1	45	rs180177110	Luminal A
	c.2101delT (p.Ser701fs)	1	43	Novel	Luminal A
*PTEN*	c.154_154+1insGG (p.His53GlyfsTer10)	1	41	Novel	Luminal A
*RAD51D*	c.680C>T (p.Ser227Leu)	1	51	rs370228071	Luminal A
*TP53*	c.818G>A (p.Arg273His)	1	48	rs28934576	TNBC
	c.1024C>T (p.Arg342Ter)	1	25	rs730882029	TNBC

HGVS: Human Genome Variation Society.

**Table 3 jpm-13-01587-t003:** List of variants of uncertain significance as detected by the American College of Medical Genetics and Genomics guideline, 2015.

Gene	HGVS	No. of Carriers	SNP ID	Molecular Subtype
*ATM*	c.8797A>G (p.Lys2933Glu)	1	rs587779875	Luminal A
	c.2944C>T (p.Arg982Cys)	1	rs587779830	Luminal A
	c.9124C>G (p.Pro3042Ala)	1	Novel	TNBC
*BARD1*	c.1174_1194delTTGCCTGAATGTTCTTCACCA (p.Leu392_Pro398del)	1	Novel	Luminal A
	c.538T>C (p.Tyr180His)	1	rs1060501311	TNBC
	c.127C>T (p.Arg43Cys)	1	rs752871324	Luminal A
*BRCA1*	c.3625T>G (p.Leu1209Val)	1	rs273900711	TNBC
*BRCA2*	c.4089C>G (p.Asn1363Lys)	1	Novel	TNBC
	c.5299_5307del (p.Lys1767_Asp1769del)	1	rs80359504	HER2-enriched
	c.5218_5223del (p.Leu1740_Ser1741del)	1	rs397507775	Luminal B
*BRIP1*	c.67C>T (p.Pro23Ser)	1	Novel	HER2-enriched
	c.1912A>G (p.Asn638Asp)	1	Novel	Luminal A
*CDH1*	c.2603G>A (p.Arg868His)	1	rs369126891	TNBC
*CHEK2*	c.1231G>A (p.Asp411Asn)	1	rs755127902	Luminal B
	c.976C>T (p.Pro326Ser)	1	rs1555917036	TNBC
*MLH1*	c.1730C>T (p.Ser577Leu)	1	rs56185292	Luminal A
	c.290A>G (p.Tyr97Cys)	1	rs773647920	Luminal A
*MSH2*	c.2435C>G (p.Thr812Ser)	1	rs1553369826	TNBC
	c.2439G>T (p.Met813Ile)	1	rs587781678	Luminal A
*MUTYH*	c.398A>G (p.Asn133Ser)	1	rs771641237	HER2-enriched
	c.158-2A>T (Splicing)	1	Novel	Luminal A
	c.892-2A>G (Splicing)	5	rs77542170	HER2-enriched * 4 Luminal B * 1
*NF1*	c.6743G>A (p.Arg2248His)	1	rs562367786	TNBC
*PMS2*	c.752T>A (p.Val251Glu)	1	Novel	Luminal A
*RAD51C*	c.722T>C (p.Val241Ala)	1	rs1555599127	TNBC
*RAD54L*	c.1862T>C (p.Leu621Pro)	1	rs768472959	Luminal A
	c.1850A>C (p.Tyr617Ser)	1	Novel	Luminal A
*TP53*	c.7G>A (p.Glu3Lys)	1	Novel	TNBC

* No of cases.

## Data Availability

The data that support the findings of this study are available on request from the corresponding author, K.K., upon request.

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
