# Peer review of "Exome Sequencing Reveals Novel Germline Variants in Breast Cancer Patients in the Southernmost Region of Thailand"

_jpm, 2023, doi:10.3390/jpm13111587_

Round 1

Reviewer 1 Report

Comments and Suggestions for Authors

In the paper entitled ” Exome Sequencing Reveals Novel Germline Variants in Breast 2 Cancer Patients in the Southernmost Region of Thailand”, the authors implemented sequencing techniques in order to identify novel variants predisposing to breast cancer development in a given population. The paper is engaging and well-written. It is important to highlight that the authors tested a relatively large group of patients (n=151), which shows a huge amount of effort put into the study. Despite the strong points of the study I mentioned, I have several minor concerns:

·         Although the aim of the study is relatively clear to the reader, it would be beneficial for the paper to clearly describe it in the introduction section

·         Please provide the list of primers used in the Sanger sequencing (e.g. in supplementary files)

·         Figure 1. (Line 173, 174) - The title and description of the figure should be placed below the graph. Also, the figure description is unclear, please rephrase it

·         Table 3 (line 180,181) – The size of the font in line 180 is different than in 181

Comments on the Quality of English Language

The paper is well-written and understandable to the reader.

Author Response

      Thank you very much for taking the time to review this manuscript. We have meticulously addressed all of the reviewer's recommendations.

Reviewer 2 Report

Comments and Suggestions for Authors

The manuscript “Exome sequencing reveals novel germline variants in breast cancer patients in the southernmost region of Thailand” is well presented and clearly written, and provides details of an exome sequencing study of 151 individuals with breast cancer. The conclusions that mutations in genes leading to susceptibility to hereditary breast/ovarian cancer syndrome are present at about 15% in this population supports the recommendation that germline genetic testing should be offered as a screening tool, much as it is in other parts of the world.

In table 1, 7 patients had DCIS according to Stage at diagnosis, but only 6 had DCIS according to the text and to Pathological group.  Which is correct?  Also in this table, it is not clear what the p values are referring to.  For instance, a p value of 0.05 is provided for the molecular subtype, but it’s not indicated which of these is statistically significant in terms of carrying PV/LP mutations.  Also, the 5th column, which is listed as % is hard to interpret- it is not the % of cases that had a PV/LP mutation but rather the % of total pathogenic mutations found. But it seems it might be more informative to provide the % of cases with a pathogenic variant- so of the 91 cases under the age of 50 at diagnosis, 19 (or 20.8%) carry a pathogenic or likely pathogenic variant.

In the text describing table 1, there is no mention of the proportion of cases with mutations of clinical relevance, but this is the most interesting part of the table.

In tables 2 and 3, it is not clear whether age means age at diagnosis or current age.  Please clarify.  Also for these tables, it would be helpful to include an extra column that specifies the tumour subtype- this information is available from supplemental data, but it is hard to tease it out for the reader and it would be interesting to present in these tables instead.

In section 3.2, line 154, the authors mention that 23 of 151 patients with BC carried “at least one cancer gene”.  In fact, every patient carries these genes.  These patients carry at least one pathogenic variant in a gene known to be associated with increased cancer risk.  This should be corrected.

In most screening programs where these genes are routinely examined for germline variants, there has been growing realization that a not insignificant proportion of the variants identified are not single nucleotide changes or small insertion/deletion events, but are larger rearrangements involving one or more exons.  There is no mention in this study whether the authors took this into account and examined their exome data for such large scale events.  While exome sequencing is not the ideal assay to identify such changes, their purported average depth of 200x should support examination of these types of variants.  It would be important to do this if it has not been done. It would be interesting if none of the patients in this cohort showed any larger rearrangements, given the frequency with which these events occur.

In the discussion (line 219) the authors maintain that 15% of patients carried a truncating germline variant.  This isn’t true- some of the variants identified were missense.

In reading a previous paper by these authors (PMID 36853301) published in 2023 in the Asian Pac J Cancer Prev, I note that many of these findings seem to have already been discussed and published, including the 48 year old individual with both an ATM and a BRCA2 variant, as well as the two individuals with the deletion BRCA1 variant, which is described in this manuscript as being novel.  It is not clear the extent to which the data presented in this paper have already been reported in the previous manuscript- this needs to be clarified.  It is confusing to read in the discussion for this submission that the BRCA1 c.1863_1885del is novel, when in fact it has already been reported by this group. In fact, in the previous manuscript (referenced above, but not referenced in the submission), all of the variants reported there are also reported here, suggesting this is an overlapping dataset.  I think this should be clarified to avoid confusion.

Comments on the Quality of English Language

English is generally good.

Author Response

(The authors gave the same response as above.)

Round 2

Reviewer 2 Report

Comments and Suggestions for Authors

Thank you for considering the points raised in my first review, and for revising the manuscript accordingly. The only additional comment would be that you might acknowledge that you are looking only at single nucleotide variants and small insertion/deletion events (ie not larger rearrangements) earlier in the manuscript- you do discuss at the end of the discussion now as a limitation, but would be good to point out the limits of your assay earlier, perhaps even in the methods as a single statement that copy number changes were not assessed.

Comments on the Quality of English Language

Some of the newly added sections do require some minor editing.

Author Response

     The reviewer's suggestions are beneficial to our work. We have done our best to edit our work based on reviewer recommendations. We thank you very much for your suggestions.
